# Biochemical and Morphological Mechanisms Underlying the Performance and Preference of Fall Armyworm (*Spodoptera frugiperda*) on Wheat and Faba Bean Plants

**DOI:** 10.3390/insects13040317

**Published:** 2022-03-23

**Authors:** Huan Liu, Yumeng Cheng, Xiaoqing Wang, Frédéric Francis, Qian Wang, Xiaobei Liu, Yong Zhang, Julian Chen

**Affiliations:** 1State Key Laboratory for Biology of Plant Diseases and Insect Pests, Institute of Plant Protection, Chinese Academy of Agricultural Sciences, Beijing 100193, China; m15210608845@163.com (H.L.); chengymcn@foxmail.com (Y.C.); wangqian_711@126.com (Q.W.); xiaobeiliu7@163.com (X.L.); 2College of Advanced Agricultural Sciences, Zhejiang Agriculture and Forest University, Hangzhou 311300, China; 201901140221@stu.zafu.edu.cn; 3Functional and Evolutionary Entomology, Gembloux Agro-BioTech, University of Liège, B-5030 Gembloux, Belgium; frederic.francis@ulg.ac.be; 4Department of Entomology, College of Plant Protection, China Agricultural University, Beijing 100193, China

**Keywords:** *Spodoptera frugiperda*, wheat, faba bean, adaptability, oviposition preference, biochemical substances, trichomes

## Abstract

**Simple Summary:**

*Spodoptera frugiperda* (J. E. Smith) has been recently identified as a notorious global crop pest that affects wheat production in China. Therefore, exploring preventive solutions based on agricultural strategies is a cost effective and eco-friendly approach in realizing sustainable pest management. The wheat–faba bean intercropping model mitigates the damage caused by wheat aphids, while the possible benefit of this pattern in the control of *S. frugiperda* remains unevaluated. To provide a fundamental basis for implementing this strategy in a wheat field for *S. frugiperda* management, this novel study attempted to extensively evaluate the effects of feeding wheat and faba bean plants on *S. frugiperda* performance and preference and to reveal the underlying mechanisms associated with the biochemical and morphological traits of the two host species. Our results suggested that the faba bean plants exhibited significant antibiosis on larvae and repellency to the females of *S. frugiperda* compared with wheat plants. Therefore, we concluded the potential usefulness of the faba bean plants as a push crop. These findings will facilitate the adoption of wheat and faba bean intercropping systems against *S. frugiperda* in the wheat-growing regions of China.

**Abstract:**

*Spodoptera frugiperda* (J. E. Smith), which attacked China in 2019, remains a significant threat to wheat production. Wheat–faba bean intercropping systems prevent damage caused by wheat aphids; however, the potential role in *S. frugiperda* control remains unclear. Here, the adaptability and preferences of *S. frugiperda* to wheat and its common intercropped plant, faba bean, were evaluated to implement an eco-friendly approach for *S. frugiperda* management. Their adaptability showed that both hosts could support *S. frugiperda* to complete their life cycle; however, the larvae performed worse on faba bean compared with on wheat. The biochemical analysis revealed that faba bean plants had lower contents of soluble sugars and total proteins but higher levels of phenolics and tannins than in wheat leaves. The gravid *S. frugiperda* preferred (during the preference assays) to oviposit on wheat rather than on faba bean plants in cage tests. The wheat odor was preferred over the faba bean odor in the Y-tube olfactometer bioassays. The morphological scanning electron microscopy (SEM) showed increased trichome density on wheat leaves. Therefore, the faba bean plants displayed antibiosis on larvae and were repellent to female moths, thus, suggesting that faba bean plants could serve as a push crop to be intercropped with wheat for *S. frugiperda* control for wheat fields.

## 1. Introduction

The fall armyworm, *Spodoptera frugiperda* (J. E. Smith, 1797), has been identified as a notorious transboundary species that originated from the tropical and subtropical regions of the Americas [1]. In January 2016, it was first detected in Africa and later spread to 44 African countries [2]. Several Asian countries continually reported the invasion of *S. frugiperda* in 2018, including India [3], Thailand, Sri Lanka, Bangladesh, Myanmar, and Yemen [4,5]. 

In January 2019, field investigations first confirmed its occurrence on corn in Pu’er, Yunnan Province, China [6]. By September 2020, this species reportedly emerged in 27 provinces (autonomous regions and municipalities) of China [7]. Generally, when 55–100% of corn plants during the mid-to-late corn development stage were infested by the devastating pest, *S. frugiperda*, it resulted in 15–73% yield loss [8], therefore, posing a significant threat to agricultural production and food security.

*Spodoptera frugiperda* larvae are known to be largely polyphagous, with more than 353 host species belonging to 76 families, with the gramineous being the most favorable species, accounting for 30.03% [9]. In many countries, this pest explicitly threatens corn production, which is the most cultivated cereal crop worldwide [10]. The research on *S. frugiperda* on corn has been broadly conducted with several approaches explored for its management [11]. Notably, the safe production of gramineous grain wheat faces the same threat posed by the greed of *S. frugiperda* as in that of corn. China is considered to be the largest wheat producer globally, and wheat production ranks third after corn (*Zea mays* L.) and rice (*Oryza sativa* L.) [12]. 

Additionally, field investigations reported that the *S. frugiperda* larvae caused severe damage to wheat in the Yunnan, Anhui, Shandong, and Henan provinces, with the damage percentage of wheat plants reaching more than 95% in some areas of the Xundian County in Yunnan Province [13]. Disturbingly, current field inspections conducted in Yunnan Province showed that its population density in wheat fields was as high as 50–60 individuals/m^2^ [14]. Several laboratory studies have proven that the *S. frugiperda* can complete their life cycle and exhibit high adaptability on wheat plants [15,16,17].

The push–pull strategy, for example (intercropping corn with leguminous crops) has been identified as an efficient and environment-friendly approach for controlling the infestation of *S. frugiperda* [18,19,20]. In Uganda, when compared with mono-cropped corn, intercropping with edible leguminous bean (*Phaseolus vulgaris* L.), soybean (*Glycine max* L.), and groundnut (*Vigna unguiculata* L.) remarkably reduced the abundance of *S. frugiperda*, especially in the early growth phases of corn [18]. Additionally, a wheat and faba bean intercropped planting pattern is widely adopted in many wheat-growing areas in China [21,22]. 

This pattern reduces nitrogen and phosphate inputs, improves diversity in the soil microbial community, improves soil and rhizosphere nutrition, and consequently increases crop yield [21,22,23,24]. Favorably, intercropping wheat with faba bean showed a noticeable reduction in the occurrence of wheat aphids [24]. However, limited information regarding wheat–faba bean intercropping in preventing *S. frugiperda* is available. Therefore, investigating the potential of faba bean intercropping in managing *S. frugiperda* under a wheat-based cropping system is a primary step.

The adaptation of insect pests to different host plants can be affected by various factors, including morphological characteristics [25,26,27], nutrients, secondary metabolites [16,28,29,30], and released volatile organic compounds [31,32,33]. The high density of leaf epicuticular trichomes is associated with plant resistance against aphids and whitefly by reducing oviposition and impeding their movement [27,34]. Generally, leaf epicuticular trichomes are divided into glandular and non-glandular trichomes according to their secretion behavior [25]. 

Compared with non-glandular trichome species, the presence of glandular ones on the leaf surfaces suppressed the settlement behavior of monarch caterpillars (*Danaus plexippus* L.) and aphids [35]. Moreover, the presence of primary and secondary metabolites, including carbohydrates, protein, and phenolic compounds in the host plant, were also recognized as factors that can determine host adaptation due to their direct influences on the absorption and use of plant nutrients by herbivores [16,28,30,36,37]. The Asian corn borer (*Ostrinia furnacalis* G.) developed better on the inbred corn line Zi330 containing high soluble sugar and protein contents [38].

Similarly, the *S. frugiperda* larvae preferred feeding on the wheat accessions with high total protein and soluble sugar contents; however, their development was inhibited after feeding on wheat accessions with high total phenol and tannin levels [16]. It was also reported that condensed phenolic tannins in cowpea (*Vigna unguiculata* L.) conferred resistance to cowpea weevil (*Callosobruchus maculatus* F.) [39]. Recently, although the physical and chemical properties involved in the host-resistant characteristics on some other phytophagous have been elucidated, few studies have analyzed the influence of these factors on *S. frugiperda* in wheat and intercropping plants, such as faba bean.

During interactions between pests and host species, the input of the insect olfactory system mediated by host plant-derived volatiles plays a crucial role [40,41,42,43,44,45,46]. These unique compounds were specific in each plant species, which can stimulate the olfactory sense of insects and induce their subsequent preference behavior [42,43]. Due to the characteristics of the rhythmic release of plant volatiles and insect-specific recognition and perception, numerous research studies have now focused on imitating and synthesizing attractants and deterrents for target pests [44,45,46]. 

The chemical substance, linalool, has been applied to monitor the oriental fruit fly (*Bactrocera dorsalis* H.) and Mediterranean fruit fly (*Ceratitis capitate* W.) attributed to its attractive effects [44,45]; it was also used to manage mosquitoes owing to its repellent function [44,46]. However, the oviposition preference of *S. frugiperda* female moths dominated by various host volatiles under the wheat–faba bean intercropping system remains unexplored.

The objective of this study was to determine whether *S. frugiperda* poses potential risks to wheat and its common intercropped plant faba bean and then compare the adaptability and preference of *S. frugiperda* to these two primary host species, which could provide insights for the integrated management of *S. frugiperda* under the wheat–faba bean intercropping system. The biochemical characteristics and physical traits of wheat and faba bean plants as well as the behavioral responses of insects to odors derived from plants were also investigated and compared in this study to elucidate the underlying mechanisms of adaptation and preference of *S. frugiperda* to wheat and faba bean plants. 

## 2. Materials and Methods

### 2.1. Insects

The larvae of *S. frugiperda* were initially collected at the end of December 2019 from a cornfield in Dehong Autonomous Prefecture, Yunnan Province, China (23°50′ N, 97°31′ E; 800 m) and reared using a slightly modified artificial diet described by Wang et al. [47] for more than 15 generations to form a stable indoor population. Once emergence occurred, the adult moths were transferred into a metal mesh cage (26 cm diameter × 30 cm height) with wet gauze and sulfuric paper as oviposition substrates, and 10% honey solution soaked in a cotton ball was provided as a food source. Egg masses on the substrates were collected daily for subsequent experiments. The population was maintained in a climate chamber (PRX-450C, Ningbo Saifu Experimental Instrument Co., Ltd., Ningbo, China) under controlled conditions at 26 °C ± 2 °C, 70% ± 5% relative humidity (RH) and 16:8 light (L):dark (D).

### 2.2. Plants

A wheat variety with resistance characteristics to *S. frugiperda*, Yannong 21 (seeds purchased in Jiangsu Aijia Ecological Agriculture Co., Ltd., Zhenjiang, Jiangsu Province, China), was screened out in the laboratory [16]. Faba bean, which is a leguminous plant that is commonly intercropped with wheat in China (seeds purchased in Zhonggeng mumin (Beijing, China) Agricultural Technology Development Co., Ltd.), were planted in plastic pots (10 cm diameter × 9 cm height) with sterile soil (mixture of nutrient matrix and loam at 3:1); they were then put into an artificial climate chamber with the same conditions as described in Section 2.1. Twenty-five-day-old wheat and twenty-day-old faba bean seedlings were used for further experimentation.

### 2.3. Adaptability Assays

The adaptability of *S. frugiperda* offspring on wheat and faba bean plants was observed using the previously described leaf segment methods [16]. First, in advance, newly formed egg masses containing about 120 eggs were collected into a Petri dish (15 cm diameter) lined with moist filter paper. Once the eggs were hatched, infancy larva were individually transferred into another new Petri dish (9 cm diameter), which contained leaf disks (3 cm length × 1 cm width) from both host plants. The daily replacement and the amount of fresh leaf disks provided depended on the larvae consumption rate in each developmental stage. The feces in the Petri dishes were cleaned regularly to prevent microbial contamination of the food sources. 

After pupation, the pupal were weighed (Sartorius BP121S, Sartorius AG, Goettingen, Germany, 0.1 mg), and the sex of the pupa was identified [48]. Then, the pupae were paired with one male and one female and put into a transparent plastic box (24 cm diameter × 11 cm height) with a cover. Wet gauze was provided on the box’s inner wall as the oviposition substrate after emergence. Meanwhile, the adult moths were offered a 10% honey solution as supplementary nutrition. Egg masses on wet gauze and box walls were collected until the adults died. The larval survival and developmental time in each stage were recorded. Each newly hatched larva was regarded as one replicate, and 100 insects were observed for each host species. The larval and pupal duration, adult longevity, and larval survival rate in each stage were calculated. All observations were carried out in a climate chamber as described in Section 2.1.

### 2.4. Plant Leaf Nutrients and Metabolites Analyses

The fresh leaves from wheat and faba bean plants were initially collected for the biochemical component assays. Then, the soluble sugar, total protein, tannin, and total phenol concentrations in fresh leaves of wheat and faba bean plants were measured, respectively, using a plant-soluble sugar detection kit, protein quantification (TP) assay kit, tannin content determination kit, and total phenol testing kit (Nanjing Jiancheng Bioengineering Institute, Nanjing, China) according to the manufacturer’s instructions. 

The soluble sugar, protein, and total phenol concentrations in the leaves were detected using a spectrophotometer (Eeymic INSTRUMENT U-T6A UV-Vis spectrophotometer, Yipu Instrument Manufacturing Co., Ltd., Shanghai, China), and the content of tannin was measured using a microplate reader (FlexStation 3 Molecular Devices, Meglu Molecular Instruments (Shanghai, China) Co., Ltd.). Four biological replicates were conducted in order to detect each biochemical substrate content. Additionally, fresh leaves were collected and weighed separately (Sartorius BP121S, Sartorius AG, Gottingen, Germany, 0.1 mg); these were further dried at 80 °C for 72 h for dry weights [48]. The calculation formula is (1):(1)$$\mathrm{Relative}\;\mathrm{water}\;\mathrm{content}=\frac{\mathrm{fresh}\;\mathrm{weight}\;(\mathrm{mg})-\mathrm{dry}\;\mathrm{weight}\;(\mathrm{mg})}{\mathrm{fresh}\;\mathrm{weight}\;(\mathrm{mg})}\times 100\%$$

In total, 20 biological replicates were performed.

### 2.5. Oviposition Preference Tests

The oviposition preference of gravid females of *S. frugiperda* to wheat and faba bean plants was measured in cage tests. First, two potted wheat and faba bean plants were arranged alternatively at the four bottom corners of the nylon cage (45 cm length × 50 cm width × 50 cm height). Then, at 8:00 pm, eight pairs (eight females and eight males) of *S. frugiperda* mated adults were introduced into the center of the cage and left to oviposit. After 72 h, egg masses deposited on each host plant and cage wall were collected. Thereafter, the quantity of newly hatched larvae was counted as the number of eggs laid on different hosts and cage walls. Finally, the tests were repeated eight times under the artificial climate chamber mentioned in Section 2.3. To avoid the bias effects caused by plant position, the relative position of the host plants was rotated between different replications.

### 2.6. Y-Tube Olfactometer Bioassays

The choice assays of gravid *S. frugiperda* females toward odors emitted by wheat and faba bean plants were observed, respectively, by dual-choice Y-tube olfactometers (20 cm stem; 20 cm arms at 75° angle; 3 cm internal diameter) as previously described with slight modifications [49]. Newly emerged *S. frugiperda* females previously spent two nights in cages containing males for mating before olfactometer assays. Specifically, odors from the two host species odor source glass jars were pumped (QC-1B air pump; Beijing Municipal Institute of Labor Protection, Beijing, China) through the two branch arms of the Y-tube olfactometers into its main stem before charcoal-purification and silica-gel moisture absorption at a flow rate of 500 mL min^−1^ through each of the branch arms. 

All the segments in this equipment were connected using Teflon tubing. Vaseline was used to seal all glass joints. Before Y-tube bioassays, the whole device was run for 30 min to avoid miscellaneous gas left in the equipment. A gravid *S. frugiperda* female pre-captured in a 50 mL centrifuge tube was individually introduced at the entry of the main olfactometer stem, and the behavior was observed for 10 min. The female walked the branch arm of either odor source for 4 cm in 10 min, which was recorded as a positive response, and those did not walk into either side of odor source or did not move into the main stem or even walk back in reverse after 10 min of introduction were identified as negative responses. 

Those who had not chosen within a given time were recorded as non-choice. The number of females selected for each plant’s odors was recorded. Five adults were set as a biological replicate, whereas 20 replicates (100 adults) were tested. The inner sides of Y-tube olfactometers and glass jars were cleaned with 75% alcohol and dried at 80 °C after each replication. Additionally, the positions of the olfactometer branch arms were reversed after changing the odor sources. All assays were performed between 20:00 and 24:00 in an entirely dark climate room at 26 °C ± 2 °C, 70% ± 5% RH.

### 2.7. Morphological Characteristics of Plant Leaves

Fresh leaves were initially cut from wheat and faba bean plants and then quickly washed in PBS buffer solution (pH 7.4) twice to remove impurities. Next, the leaf samples were fixed with 4% glutaraldehyde for 48 h and washed with 0.1 mol L^−1^ PBS buffer, further fixed with 1% osmium tetroxide (OsO_4_) for 2 h, and washed with distilled water for 30 min. 

Next, the samples were dehydrated in a graded ethanol series (50–100%), placed in iso-amyl acetate for another 30 min, and then air-dried at a critical point using Baltec CPD-030 critical point dryer (Bal-Tec, Zurich, Switzerland). Samples were a sputter-coated with a 10 nm layer of gold with ion-sputtering equipment (Hitachi IB-5, Eiko, Tokoy, Japan). After that, images were observed using a scanning electron microscope (SEM, Hitachi-SU8010, Hitachi, Tokoy, Japan), and three fields of view were captured for each leaf sample for both adaxial and abaxial surfaces. The microscope then automatically generated scales for each image at the time of visualization. For a selected leaf surface, the trichome density per cm^2^ was counted manually. At least ten images were captured from the adaxial and abaxial sides for each host type [50].

### 2.8. Statistical Analyses

The Shapiro–Wilk and Levene’s tests in one-way analysis of variance were adopted to verify the normality and homogeneity of variance, respectively. The developmental duration in each stage, the adult longevity and the pupal weight of *S. frugiperda* reared on wheat and faba bean leaves, the biochemical contents of host leaves, and the trichome density on adaxial or abaxial sides of both hosts were evaluated using independent sample *t*-tests. The oviposition preference of *S. frugiperda* female moths to wheat and faba bean plants in cage tests was compared via Tukey’s HSD test. 

The nonparametric Kruskal–Wallis tests followed by multiple comparisons were used to estimate the preference differences of *S. frugiperda* female adults to two host odor sources in Y-tube olfactometer assays. The percentages were arcsine square-root-transformed to homogenize variances before analysis. *p*-values less than 0.05 were considered statistically significant. Data analysis was performed using IBM-SPSS v.19.0 (SPSS Inc., Chicago, IL, USA) for Windows, and the figures were drawn in GraphPad Prism version 8.0 (GraphPad Software, Inc., San Diego, CA, USA).

## 3. Results

### 3.1. Developmental Duration of S. frugiperda on Wheat and Faba Bean Leaves

As shown in Table 1, *S. frugiperda* completed their life history on wheat and faba bean leaves. However, the developmental duration for each instar larvae of *S. frugiperda* feeding on faba bean leaves was significantly longer than on wheat leaves (all *p*-values < 0.001), and no significant difference was found in the pupal duration (*p* = 0.7244) and adult longevity (*p* = 0.743).

### 3.2. Survival Rate of S. frugiperda Larvae on Wheat and Faba Bean Leaves

Different host species can significantly affect the curves of the stage-specific survival rate (*s_xj_*) of larvae and the probability shows that newly hatched larvae survive to age *x* and develop to stage *j*. The probability that a newly hatched neonate of *S. frugiperda* surviving up to the adult stage differed remarkably between wheat (Figure 1A) and faba bean (Figure 1B) leaves with 80% on wheat and only 16% on faba bean. The highest mortality rates were observed to occur in the first and second instar larvae with 46% and 76% (Figure 1B) after feeding on faba bean, which were significantly higher than the results on wheat leaves (1% and 5%), respectively (Figure 1A).

### 3.3. Pupal Weight of S. frugiperda after Reared on Wheat and Faba Bean Leaves

No significant difference was observed in terms of the pupal weight after the *S. frugiperda* larvae were reared on wheat and faba bean leaves (*t* = 1.686, *df* = 29.465, *p* = 0.102, Figure 2). The pupal weight of *S. frugiperda* reared on wheat leaves was between 110.4 and 267.1 mg, whereas they ranged from 99.5 to 238.7 mg on faba bean leaves.

### 3.4. Contents of Biochemical Components in Wheat and Faba Bean Leaves

The contents of various biochemical components in wheat and faba bean leaves, including the relative water content, soluble sugar, total protein, total phenol, and tannin contents, were measured in this study. No significant difference was noted in the relative water content (*t* = 0.552, *df* = 19.895, *p* = 0.587, Figure 3A) and total phenols (*t* = −2.365, *df* = 4, *p* = 0.077, Figure 3D) between these two host leaves. However, the contents of soluble sugars (Figure 3B), total protein (Figure 3C), and tannins (Figure 3E) were found to differ significantly between the two host species. Specifically, the soluble sugars (*t* = 5.829, *df* = 4, *p* = 0.004) and total protein content (*t* = 10.13, *df* = 4, *p* = 0.001) in wheat leaves were significantly higher than in faba bean leaves, whereas the tannin content was significantly lower than in faba bean leaves (*t* = −72.121, *df* = 2.25, *p* < 0.001).

### 3.5. Oviposition Preference of S. frugiperda Female Moths to Host Species

The choice assays in cage tests showed that the oviposition preference of *S. frugiperda* female moths to wheat and faba bean plants was significantly different (F = 6.014, *df* = 2, 23, *p* = 0.009, Figure 4). Additionally, the number of eggs laid on faba bean plants was significantly lower (by approximately 2.7 times) than on wheat plants. However, the female moths deposited some eggs on the cage wall with no significant difference in the number of eggs deposited on the cage wall compared to wheat (*p* = 0.112) and faba bean plants (*p* = 0.397).

### 3.6. Preference of S. frugiperda Female Moths to Volatiles Emitted from Two Host Species

Significant differences in the orientation preference of *S. frugiperda* female adults to the odors from wheat and faba bean plants, respectively, were observed (χ^2^ = 21.556, *df* = 2, *p* < 0.001, Figure 5). Additionally, the mated females significantly preferred the odors from wheat plants compared to faba bean plants (*p* < 0.0001).

### 3.7. Observation of Trichome Types and Density on Wheat and Faba Bean Leaves

As per the SEM results, we found that the trichome types on wheat and faba bean leaf surfaces were significantly different. The trichome types on the wheat leaves were noted to be bristle (Figure 6A,B); however, there were hooked glandular hairs on the leaf surface of faba bean leaves (Figure 6C,D). Significant differences were observed in terms of trichome density between the two host plant species (*p* < 0.001, Figure 7). Few trichomes were observed on both sides of faba bean leaves. Additionally, the trichome density on both adaxial (*t* = −10.215, *df* = 5, *p* < 0.001) and abaxial sides (*t* = −14.192, *df* = 5, *p* < 0.001) of wheat leaves was significantly higher than that of faba bean leaves.

## 4. Discussion

The adaptability of herbivorous insects has been demonstrated to be affected by host species [49], mainly due to differences in plant morphologies [25,26,27], palatability [49], nutritional contents, and secondary metabolites [16,28,29,30,36,37]. Several studies have reported that various host species substantially affected the development and reproduction of herbivorous pests, such as *S. frugiperda,* and that gramineous crops, such as corn and wheat, were among the most preferred food sources [9,10,15,20].

In wheat cropping system, the wheat-leguminous plant intercropping system pattern was usually presented as a push–pull strategy to control aphids [23,51,52]. However, the efficacy of this strategy in an intercropping system in which faba bean is used as the push crop to manage *S. frugiperda* in wheat fields remains unclear. Therefore, we compared the performance of *S. frugiperda* between wheat and its common intercropped crop, faba bean. Our results showed that wheat and faba bean plants supported *S. frugiperda* in completing their life cycle. The development duration of larvae feeding on faba bean leaves was significantly longer than that of larvae feeding on wheat leaves. 

Moreover, the survival rate of larvae feeding on wheat was significantly higher than that of those feeding on faba bean leaves. These results show that faba bean is a less suitable host species compared with wheat. Previous studies have documented that the larval diet can affect the development and reproduction of *S. frugiperda* [10,15,49,53], with research indicating that the larvae of this pest developed better on plants belonging to the family Gramineae but performed poorly on other hosts, including soybean, tomato (*Solanum tuberosum* L.), bean (*Phaseolus vulgaris* L.), and desmodium (*Desmodium intortum* (Mill) Urb.) [15,20]. 

Similarly, another recent study observed a significantly lower survival rate and longer mean generation time of *S. frugiperda* larvae fed on soybean rather than wheat [15]. Our findings also confirmed that *S. frugiperda* showed poor adaptability on faba bean plants compared with wheat, suggesting that faba bean plants can be used as potential repellent crops in a wheat planting system to control *S. frugiperda*.

The nutrient compounds and secondary metabolites in host plants are vital biochemical characteristics that determine insect adaptability. The performance of the insect pest longicorn (*Anoplophora chinensis* E.) correlated positively with the soluble sugar content in *Casuarina* spp. leaves but was negatively correlated with the total phenol, tannin, and flavonoid contents [54]. *O. furnacalis* developed better on the corn inbred line Zi330, which has high soluble sugar and protein contents [38]. In this study, the soluble sugar and total protein contents were significantly higher in wheat leaves compared with in faba bean leaves, indicating that faba bean has poor nutritional value for *S. frugiperda* larvae in comparison with wheat. 

Phenolic compounds are considered major secondary metabolites against herbivores in various plant species [36,54,55,56] with defensive tannins mainly serving as feeding deterrents against several phytophagous pests, such as gypsy moth (*Lymantria dispar* L.), brown-tail moth (*Euproctis chrysorrhoea* L.), *O. brumata*, and cowpea aphids. (*Aphis craccivora* K.), decreasing the quality of plant tissue by binding to salivary proteins and inducing strong astringency in the mouth [36]. In *Arabidopsis*, the presence of tannin and glucosinolates promoted the resistance against cabbage aphid (*Brevicoryne brassicae* L.) and mustard aphid (*Lipaphis erysimi* K.) [57,58]. 

Although there was no noticeable difference in terms of the total phenol content between wheat and faba bean leaves in the present study, the tannin content was significantly higher (6.19 times) in faba bean leaves compared with in wheat leaves. Our combined analysis of the biological parameters of *S. frugiperda* on faba bean leaves suggested that lower nutritional quality and higher toxic tannin content in faba bean leaves resulted in the low adaptability of *S. frugiperda* larvae. Investigations of the typical biochemical substances in plant species revealed the resistant mechanisms of host species and can also encourage chemical ecology studies [36]. Therefore, further functional verification of secondary plant substances in faba bean plants is warranted.

The adaptability of phytophagous insects on different host plants initially depends on the oviposition preference of female adults due to their limited mobility during the larval stage [10,16]. In cage tests, several eggs were oviposited on wheat plants, which were higher in number than those oviposited on faba bean plants, thus, indicating that the faba bean plants had repelling effects on the oviposition of *S. frugiperda* female moths. This can be attributed to the genetic traits of *S. frugiperda*’s preference for gramineous plants [9,10,15,20]. 

Additionally, the current results are consistent with the “mother knows best” theory, which suggests that female moths lay their eggs on the most suitable host plants for the development of their offspring and the expansion of the population size [16,30]. Specifically, the female moth would not lay many egg masses on faba bean plants under selective conditions that are not suitable for establishing their population.

Plant-derived volatiles have been proven to affect the oviposition preference of phytophagous pests [31,32,33]. In Y-tube olfactometer assays, the mated females significantly preferred the odors from wheat plants compared with those from faba bean plants. A previous study reported that the black bean aphid (*Aphis fabae* S.) responded behaviorally to a 15-component blend of electrophysiologically active volatile compounds emanating from faba bean plants with octanal, (*R*)-(-)-linalool, and (*S*)-(-)-germacrene D showing repellent activity to *A. fabae* [59]. However, the specific volatile components and their role in attracting and repelling *S. frugiperda* in wheat and faba bean plants remains unknown; thus, further analysis should be performed via gas chromatography–mass spectrometry (GC-MS). 

Notably, Lepidopteran larvae, despite their limited mobility, can still rely on their immature sensory system to realize the host orientation using plant volatiles as cues to orient the host plants or away from non-host plants. Neonate larvae of the European corn borer (*Ostrinia nubilalis* Hübner) are able to adapt to the odors from corn plants while avoiding the odors from spinach (*Spinacia oleracea* L.) plants [60]. This is partly related to the nutritional value provided to the insects by the host plant as spinach produces (non-volatile) phytosterols that are toxic and deterrent [60] and partly related to the rhythmic release of plant volatiles and plant odors elicit different behavioral responses, either repellency or attraction, in neonate larvae of European corn borer [61]. 

Therefore, the role of host plant volatiles in the orientation of *S. frugiperda* newborn larvae to host species, wheat and faba bean plants, needs to be similarly demonstrated in the future. Additionally, several researchers reported that the morphological properties of host species had a significant influence on the oviposition preference of phytophagous insects [25,26,27,62]. As a morphological barrier on the plant surface, trichomes essentially act as a host defense mechanism during the selection of potential oviposition sites [63].

The morphological SEM observations revealed that the trichome types and densities on wheat and faba bean leaf surface were significantly different; the trichome density on wheat leaves was remarkably higher compared with on faba bean leaves. Some research has shown that the presence of trichomes on host leaves reduced larval mobility and adult oviposition preference [27,35]. On the other hand, the trichome density or length on tomato, soybean, and wormseed plants negatively affected the oviposition of cabbage looper, *Trichoplusia ni*, females [27]. 

Contrary to these findings, we proved that wheat plants with a higher density of trichomes attracted more *S. frugiperda* females compared with faba bean plants with a lower density of trichomes. It has been speculated that an appropriately high density of trichomes is necessary for *S. frugiperda* moths to hold firmly to avoid falling and oviposit eggs [64]. Additionally, the differences in the volatile compounds emitted from wheat and faba bean plants may be another critical factor affecting the oviposition preference of *S. frugiperda* females, thus, shielding the effects of the structure and density of trichomes.

## 5. Conclusions

According to our findings, we determined that the larvae of *S. frugiperda* performed poorly and that female moths preferred faba bean plants for oviposition when compared to wheat, which suggests that the faba bean was not deemed suitable for *S. frugiperda* development. Additional analysis indicated that the poor performance and non-preference of faba bean plants by *S. frugiperda* were attributed to the lower contents of nutritional compounds and lower low trichome density but higher tannin levels and varying plant-derived volatiles in leaves. Conclusively, this study established that faba bean is a promising repellent crop for *S. frugiperda*, which may not be favorable when adopting faba bean as a push crop to mitigate *S. frugiperda* damage in wheat–faba bean intercropping systems.

## Figures and Tables

**Figure 1 insects-13-00317-f001:**
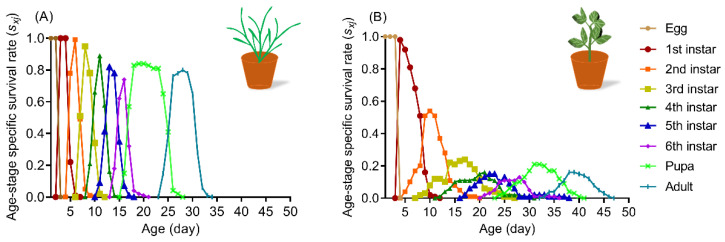
The age-stage-specific survival rate (*s_xj_*) of *S. frugiperda* reared on wheat (**A**) and faba bean (**B**) leaves.

**Figure 2 insects-13-00317-f002:**
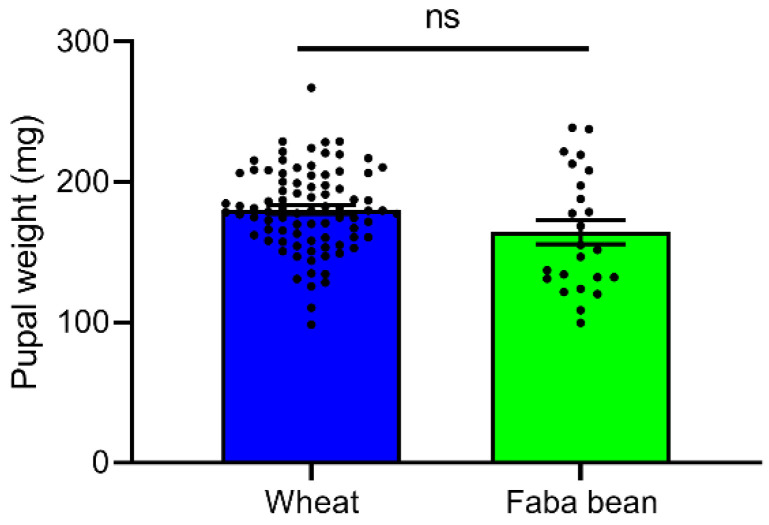
The pupal weight of *S. frugiperda* reared on wheat and faba bean leaves. All data are presented as the mean ± standard error (SE). The value bars with “ns” are insignificant (independent sample *t*-tests, *p* > 0.05).

**Figure 3 insects-13-00317-f003:**
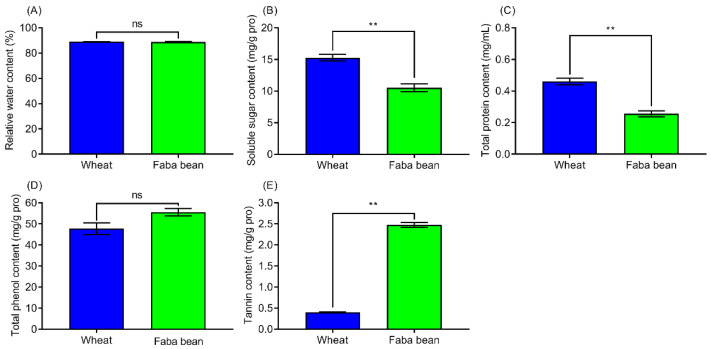
The relative water (**A**), soluble sugar (**B**), total protein (**C**), total phenol (**D**), and tannin contents (**E**) in wheat and faba bean leaves. All data are presented as the mean ± standard error (SE). The value bars with “ns” are not significant; “**” on the top of standard error bars indicate a significant difference between two host species (independent sample *t*-tests, *p* < 0.01).

**Figure 4 insects-13-00317-f004:**
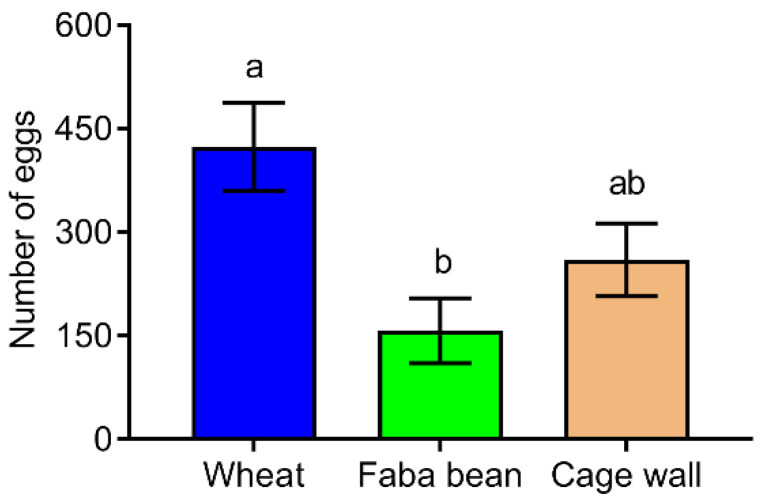
The oviposition preference of *S. frugiperda* for wheat and faba bean plants in cage tests. All data are presented as the mean ± standard error (SE). Different letters on the top of standard error bars indicate significant differences among groups (Tukey’s HSD tests, *p* < 0.05).

**Figure 5 insects-13-00317-f005:**
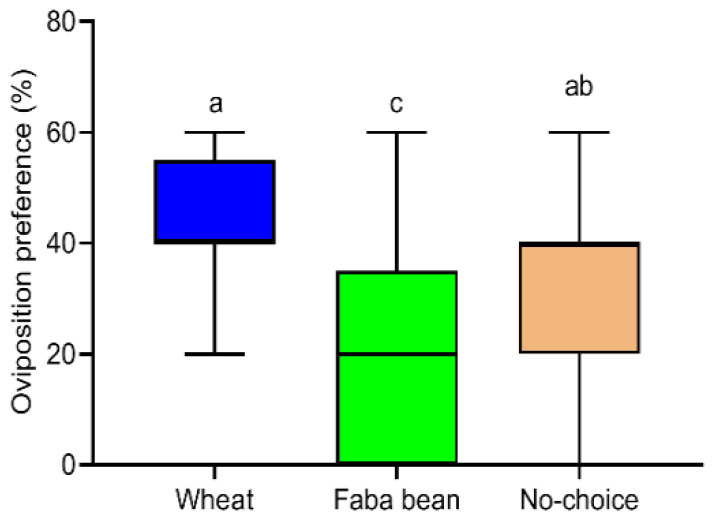
The median preference of *S. frugiperda* female moths to the odors emitted from wheat and faba bean plants in Y-tube olfactometer tests. Different letters above the bars indicate significant differences among groups (nonparametric Kruskal–Wallis tests, *p* < 0.05).

**Figure 6 insects-13-00317-f006:**
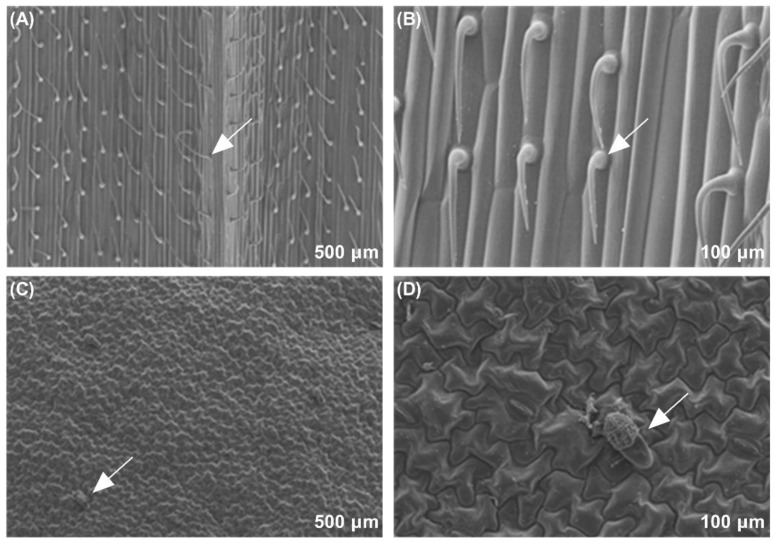
Scanning electron microscope (SEM) observations of the epidermal characteristics of wheat and faba bean leaves. The adaxial and abaxial surfaces of wheat leaves (**A**,**B**) and adaxial and abaxial surfaces of faba bean leaves (**C**,**D**). The arrow in the figures indicates the shape of the trichomes in the field of view.

**Figure 7 insects-13-00317-f007:**
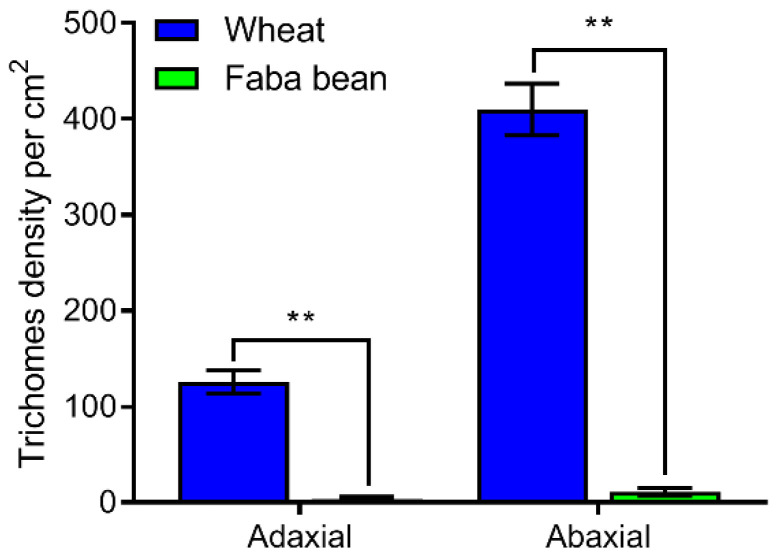
The trichome density (per cm^2^) of wheat and faba bean leaves. All data are presented as the mean ± standard error (SE). “**” on the top of standard error bars indicate a significant difference between two host species (independent sample *t*-tests, *p* < 0.01).

**Table 1 insects-13-00317-t001:** The development duration and adult longevity of the *S. frugiperda* reared on wheat and faba bean leaves.

Stage	Wheat		Faba Bean		Significant
	*n*	Days	*n*	Days	
Egg	100	2.00 ± 0.00	100	2.00 ± 0.00	ns
Larva					
1st instar	100	2.23 ± 0.045	100	5.09 ± 0.16	***
2nd instar	100	2.32 ± 0.053	69	4.32 ± 0.21	***
3rd instar	100	2.60 ± 0.065	39	5.31 ± 0.29	***
4th instar	99	2.54 ± 0.056	34	4.03 ± 0.21	***
5th instar	99	2.56 ± 0.071	30	4.03 ± 0.23	***
6th instar	96	2.01 ± 0.060	27	3.22 ± 0.26	***
Pupa	86	7.91 ± 0.13	25	7.40 ± 0.40	ns
Adult	80	5.83 ± 0.10	20	5.90 ± 0.25	ns

All data are presented as the mean ± standard error (SE). “*n*” represents the number of samples in each developmental stage. The “***” or “ns” in the rightmost column indicate that there are significant (independent sample *t*-tests, *p* < 0.001) or no significant differences (*p* > 0.05) between two host species.

## Data Availability

Data can be provided upon request from the lead author.

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
