# Peer review of "Biochemical and Morphological Mechanisms Underlying the Performance and Preference of Fall Armyworm (Spodoptera frugiperda) on Wheat and Faba Bean Plants"

_insects, 2022, doi:10.3390/insects13040317_

Round 1

Reviewer 1 Report

Review ID 1627331

Biochemical and morphological mechanisms underlying effects 2 of feeding wheat and faba bean plants on the performance and 3 preference of the fall armyworm (Spodoptera frugiperda)

               In present time, environmentally friendly methods of plant protection are of importance where natural defense mechanism of plants based on volatile organic compounds (VOCs) may play significant role.

               This is quite well organized manuscript. I found this “ms” interesting and innovative. However, a few questions must be explained more precisely.

Critical review:

  1. Lines 122-128. The aim of the study is poorly presented. Try to be more precise, please.
  2. Lines 132-133. Geographical coordinates, please.
  3. Lines 199-200. What does it mean 48-h-old? Were they expose to the food or kept in starvation? It is very importnat for the reaction of insects.
  4. Lines 205-206. It is strange. Does it mean that after 30 min run everything was fixed by this run?
  5. frugiperda and Y-tube olfactometer? This is not adequate method for the moths observation.
  6. I don't see the odor releasing way to the insects. It is not described precisely.
  7. What about odors? Which odors are responsible for the insect reaction?
  8. Lines 425-434. What about odors? No conclusions connected to odors?
  9. Discussion in present form is rather very poor and not constructive. I suggest to add a few additional references.

Some other papers of usefulness for Discussion:

Orientation of European corn borer first instar larvae to synthetic green leaf volatiles

Journal of Applied Entomology 137(3), 234-240 (2013)

DOI: 10.1111/J.1439-0418.2012.01719.X

Pulsed Odors from Maize or Spinach Elicit Orientation in European Corn Borer Neonate Larvae

Journal of Chemical Ecology 35, 1032–1042 (2009)

DOI: 10.1007/S10886-009-9676-7

Reviewer 2 Report

The manuscript insects-1627331- titled “Biochemical and morphological mechanisms underlying effects of feeding wheat and faba bean plants on performance and preference of the fall armyworm (Spodoptera frugiperda) is an interesting article which highlights the possible integration of faba bean as a push-crop in a push-pull strategy for FAW management in a wheat field. This study highlights an emerging crisis in the management of FAW in wheat fields, and potential FAW damage when a poor choice of legume is used for wheat intercropping, which is interesting for the scientific audience.

GENERAL COMMENTS

The study concludes that lower content of soluble sugars, lower total proteins and high levels of tannins and phenolic compounds in faba bean are the deterrents of FAW, suggesting that faba bean plants could be served as a push crop to be intercropped with wheat for S. frugiperda control on a wheat field (Lines 41-42). However, this conclusive inference that faba bean could be intercropped with wheat to control FAW in wheat fields is not convincing. For instance, if faba bean is considered as a push crop while wheat is a favourable FAW host as demonstrated by this study, then intercropping wheat with faba bean could be devastating to wheat fields since the faba bean will push the FAW to attack more wheat crops. This situation be can only be favourable where faba bean is the farmer’s preferred crop. I think the conclusion should rather be that faba bean is not a favourable intercrop for wheat if potential FAW attack is considered. A favourable intercrop should pull the FAW away from the main crop to the secondary crop, not pushing FAW away from the secondary crop into the main crop. Would be good to check the silicon content in the plant material, which is known to influence pest incidence and damage.

  • The title is long and a little confusing. Consider revising title to “Biochemical and morphological mechanisms underlying the performance and preference of fall armyworm (Spodoptera frugiperda) on wheat and faba bean plants”.
  • I am not sure of the requirements of this Journal “Insects” but nine keywords seem too many
  • 72 references also seem too many for this manuscript
  • There are some lengthy sentences in the manuscript that obscure the reading. The authors should keep all sentences short and straight to the point.
  • The methodology does not explain under what conditions the maize and faba bean were cultivated. The agronomic practices employed can affect the behaviour (i.e., adaptability and preference) of any pest. However, I assume that this study was conducted only in controlled lab/climate chamber conditions. If so, why are the authors giving the impression of a wheat-faba bean intercropping study, which is not the case. It is simply a controlled preference and adaptability study for the two crops, which could be different under field conditions especially when they are intercropped.

POINT-BY-POINT COMMENTS

Line 86: “Interplanting” or intercropping?

Line 87: “Planting ecosystem” or cropping system?

Lines 122-124: This is unnecessarily long “Therefore, to compare the adaptability and preference of S. frugiperda to wheat and its common intercropped plant, faba bean, elucidating its mechanisms, this study aimed to observe the performance of S. frugiperda on these two primary host species”. Consider revising this sentence to “This study aimed to elucidate the mechanisms adaptation and preference of S. frugiperda on wheat and its common intercropped plant, faba bean

Lines 124-128: These two sentences reflect additions to the aim of this study, which I think is unnecessary. The authors can replace these sentences with well formulated hypothesis that would guide this study. 

Lines: 132-135: “Colony kept for more than 15 generations”.

  • How did the authors decide on the 15 generations?
  • There is no mention of whether field populations were added intermittently in order to reduce genetic degradation.
  • The dimension of the leaf disc should be specified. Also give a rough estimate of the number of leaf disc inserted. “Enough leaf disc” is not sufficient.

Line 151: Section 2.3 and Line 171: Section 2.4 have the same title: “Adaptability assays”. Please review this and consider alternative names.

Lines 157-159: It is good to indicate that your replacement frequency and number of fresh leaf disc provided depended on the larvae consumption rate, but you should please give the actual replacement frequency that you executed (e.g., daily or every two days, etc.) and the exact number of fresh leaves provided.

Line 161: one male and one female pupae were kept in transparent plastic boxes.

  • Did the authors pay attention to ages of the pupae? This is because male and female pupae of the same age do not become adult at the same time.

Line 189: Ages of the two potted wheat and bean plants were not mentioned. It will be important to know the ages of the plants that were used.

Line 191: How did authors decided to use eight pairs of males and female?

Line 191: You now mentioned 8 PM but in the previous sentence you did not mention when the experiment started. So why 8 PM?

Line 210: Specify the exact time after which the gravid females were recorded as non-choice. To say ‘after a given time’ is not specific.

Line 218: pH7.4 and not PH7.4

Table 1: It will be good for the authors to add on the table, the total developmental time on both wheat and faba bean

Lines 264 - 266: “80.00%”, “46.00%”, “76.00%” – consider revising throughout the MS to “80%, 46%, 76%

Line 265: Grammar: sentence should read “the highest mortality was observed in the 1st and 2nd instar larvae…

Line 273 - 274: “reared by” appears twice in the sentence. These should be revised throughout the manuscript to “reared on”

Line 297: Revise to “The choice assays in cage tests showed that oviposition preference…”

Lines 299 - 300: Revise to “Additionally, the number of eggs laid on faba bean plants was significantly lower (approximately 2.7 times) than on wheat plants”.

Lines 303 – 307: Figure 4. Authors should review the statistic here. From the graph. The s.e bars of wheat and cages wall clearly do not touch each other, implying that they should be significantly different (P < 0.05).

Lines 311 - 312: Revise to “Additionally, the mated females significantly preferred the odors from wheat compared to faba bean plants (P < 0.0001)”.

Lines 348 - 349: Revise this sentence to reflect the discussion not presentation of results as it is now.

Lines 356 - 357: “lower survival rate and longer mean generation time of S. frugiperda larvae fed on the soybean than wheat [16]”

Lines 358 - 424: The discussion is logical and related to the results obtained in this study but there are some minor grammatical errors to be corrected. I strongly recommend language editing if the authors are not native English speakers.

Lines 427 - 429: Revise this sentence to “and female moths preferred faba bean plants for oviposition when compared to wheat, which suggests that the faba bean was not deemed suitable for S. 428 frugiperda development”.

Lines 429 - 432: Revise this sentence to “Additional analysis indicated that the poor performance and non-preference of faba bean plants by S. frugiperda were attributed to the lower contents of nutritional compounds and low trichome density, but higher tannin levels and varying plant-derived volatiles in leaves”.

Lines 432 - 434: Revise this sentence to “Conclusively, this study has established that faba bean is a promising repellent crop for S. frugiperda, which is a potential that that may not be favourable when adopting faba bean as a push crop to mitigate S. frugiperda damage in wheat-faba bean intercropping systems”.

Lines 432 - 434: The authors should reconsider their view of using a good push crop as an ideal intercrop plant. Where is the push plant pushing the pest to? Likely towards the wheat as main crop, which could be catastrophic for the wheat plants! My understanding is that the proponents of wheat-faba bean have focused on the nutritional benefits of this intercrop without due consideration to the aspects like emerging pests. Therefore, under such considerations of these emerging pests, intercropping these two plants may not be the best option where FAW damage is in perspective. I suggest that the authors should consider taking these finding into another study in actual field intercrop situation.

Reviewer 3 Report

This is a good paper well written and elaborate, which deserves to be accepted for publication. I am just suggesting to the authors to correct a tipo error at line 131 the title “Inscets” instead of “Insects”.

Alos, why using the same title? at Lines 151 and 170. I suggest to use 2 different titles: For example line 151 refers mostly to Larval development assays on plants and the line 170 to Plant metabolites analyses. Then for line 216, I would suggest better Plant leaves morphological characterizations or something like that.

Round 2

Reviewer 1 Report

Significant corrections were attached.